# GBO: A Multi-Granularity Optimization Algorithm via Granular-ball for Continuous Problems

## Abstract

Optimization problems aim to find the optimal solution, which is becoming increasingly complex and difficult to solve. Traditional evolutionary optimization methods always overlook the granular characteristics of solution space. In the real scenario of numerous optimizations, the solution space is typically partitioned into sub-regions characterized by varying degree distributions. These sub-regions present different granularity characteristics at search potential and difficulty. Considering the granular characteristics of the solution space, the number of coarse-grained regions is smaller than the number of points, so the calculation is more efficient. On the other hand, coarse-grained characteristics are not easily affected by fine-grained sample points, so the calculation is more robust. To this end, this paper proposes a new multi-granularity evolutionary optimization method, namely the Granular-ball Optimization (GBO) algorithm, which characterizes and searches the solution space from coarse to fine. Specifically, using granular-balls instead of traditional points for optimization increases the diversity and robustness of the random search process. At the same time, the search range in different iteration processes is limited by the radius of granular-balls, covering the solution space from large to small. The mechanism of granular-ball splitting is applied to continuously split and evolve the large granular-balls into smaller ones for refining the solution space. Extensive experiments on commonly used benchmarks have shown that GBO outperforms popular and advanced evolutionary algorithms. The code is available in the Supplementary Materials.

## 1 Introduction

Optimization is a key research area in science and engineering, focused on identifying optimal solutions Molaei et al. (2021). It spans various fields, including engineering design Liu et al. (2012); Saha et al. (2021); He et al. (2023), gene recognition Xu et al. (2022), traffic signal control Bi et al. (2014); Li & Sun (2018), machine learning Barshandeh et al. (2022); Abdollahzadeh et al. (2024); Li et al. (2023), and medical issues Lian et al. (2024), among others.

Early works focused on deterministic search methods such as gradient descent Tsitsiklis et al. (1986); Ruder (2016), Newton's method Fischer (1992), mixed integer programming Shen et al. (2023), etc. These methods usually require mathematical calculations and are prone to getting stuck in local optima. In large-scale environments, the solution space of optimization problems grows exponentially, making such methods no longer effective.

Evolutionary optimization, inspired by natural evolution and biological behavior, has increasingly been applied to algorithm design and complex problem-solving. Representative methods include genetic algorithm, particle swarm algorithm, ant colony algorithm Holland (1992); Kennedy & Eberhart (1995); Dorigo et al. (2006), etc. These methods iteratively and randomly search for the optimal solution through mutual learning and competition among individuals in the population. It does not rely on strict mathematical models, and can effectively handle complex optimization characteristics in big data environments.

Despite providing high-quality solutions to complex problems and attracting significant research interest, heuristic optimization methods often overlook the granular characteristics of different regions within the solution space (as shown in Figure 1). As illustrated in Figure 1, the contours delineate various regions within the solution space, each exhibiting distinct levels of granularity. For instance, regions proximate to the global optimum are characterized by finer granularity, whereas regions farther from the optimum display coarser granularity. However, not all regions have the same optimal solution potential.

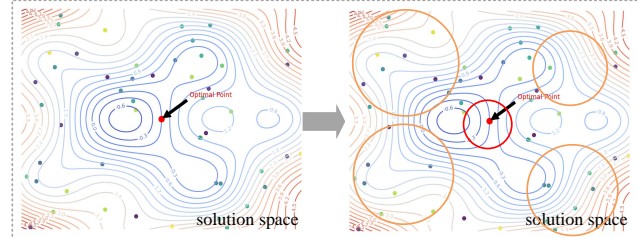

Figure 1: A schematic of different granularity characteristics in the solution space.

Modeling the granular characteristics of the solution space is not easy. Because there are several challenges: **(1)** The primary challenge is how to effectively characterize the granular characteristics of the solution space. Existing studies typically employ fine individual granularity to search the entire solution space, often neglecting its granular characteristics. This approach fails to address the complexity and diversity present in different regions effectively. **(2)** The second challenge lies in accurately assessing the potential optimality within each region. In the solution space, different regions may harbor varying degrees of optimal solutions, and traditional methods often struggle to precisely identify and evaluate these potentially optimal regions. Thus, it is crucial to develop an optimization algorithm capable of capturing the connections and differences between sub-regions of the solution space from a multi-granularity perspective.

To this end, we propose a multi-granularity optimization algorithm via granular-ball (GBO) for solving complex continuous optimization problems. Specifically, Multi-Granularity Solution Space Refinement involves initially covering the entire solution space with a coarse-grained granular-ball and then using a splitting mechanism to split fine-grained child granular-balls. Furthermore, Granular-ball Exploration and Exploitation involves the collaborative search among multiple child granular-balls. The coarse-to-fine search process better exploits potential differences in optimal solutions across various regions. These mechanisms replace the traditional point-based iterative search with a regional search approach, allowing for a more comprehensive consideration of the complexity and distinctiveness of the solution space. Experiments on benchmark and real-world problems show that GBO surpasses the classic and popular algorithms. Our contributions are summarized as follows:

- We highlight the unique granular characteristics of sub-regions within the solution space and investigate the potential of granular-balls in solving complex continuous optimization problems.

- We propose a multi-granularity optimization algorithm via granular-ball (GBO). This method characterizes the solution space from coarse-grained to fine-grained through two stages, namely, coarse-grained granular-ball initialization and fine-grained granular-ball splitting. Our method compensates for the drawbacks of potential large optimal region deviations that may arise from single-point searches by utilizing set-based searches.

- We present a granular-ball exploration and exploitation process that includes the generation of guiding granular-balls and the elite granular-ball retention operations. Through the multi-granularity characterization and search of the solution space, the evolutionary mechanism based on population and random search is maintained.

- We conduct extensive experiments on benchmark Liang et al. (2013) and real-world optimization problem. The results verify the efficiency and accuracy of GBO in solving complex continuous optimization problems.

## 2 RELATED WORK

**Evolutionary Algorithms (EAs).** Evolutionary Algorithms are global optimization techniques inspired by natural evolution, commonly used for complex problems. They are population-based methods that iteratively improve solutions towards optimality Jin & Branke (2005). The earliest

evolutionary algorithms were directly inspired by biological evolution processes, forming the foundation of evolutionary computation through the simulation of natural selection and genetic variation. A classic example is the Genetic Algorithm (GA) Holland (1992), which simulates biological genetic evolution by using selection, crossover, and mutation operations to enhance individual fitness. Later, Evolution Strategies (ES) Beyer & Schwefel (2002) were developed to focus on optimizing continuous variables and improving efficiency through adaptive mutation strength adjustments. These algorithms not only focus on individual evolution but also emphasize collaboration and information sharing among populations. Ant Colony Optimization (ACO) Dorigo et al. (2006) is an example that imitates the pheromone-based path selection process of ants during foraging, making it particularly suitable for combinatorial optimization problems like the Traveling Salesman Problem. Particle Swarm Optimization (PSO) Kennedy & Eberhart (1995) simulates the behavior of bird flocks searching for food, utilizing information exchange between individuals and their neighbors to achieve dynamic optimization.

Evolutionary algorithms have also been successfully employed in combinatorial optimization problems. Specifically, Xiang et al. (2019) proposed a PSO strategy (PBS-PSO) based on proportional integral differentiation (PID), which takes advantage of past, current, and global best changes to update the search direction to accelerate convergence and adjust the search direction to get rid of local optima. Zhang et al. (2018) proposed a multi-objective particle swarm optimizer based on a competitive mechanism, in which the particles are updated based on pair competitions performed in the current swarm in each generation.

Overall, evolutionary algorithms offer diverse tools and methods for solving complex optimization problems by simulating various natural evolutionary and adaptive mechanisms. Their flexibility and adaptability make them crucial for a wide range of real-world problems, and they continue to advance in research and engineering applications.

**Granular-ball Computing (GBC).** Chen (1982) pointed out that human cognition has the law of "global precedence" in his research published in Science. Based on the theoretical basis of traditional granular computing, Wang (2017) took the lead in proposing multi-granular cognitive computing in combination with the cognitive law in human brain cognition. Xia et al. (2023) introduced an innovative computational method known as granular-ball computing (GBC), celebrated for its efficiency and robustness.

The reason for Xia et al. (2023)'s approach to multi-granularity feature representation using granular-ball is that the geometry of a granular-ball is completely symmetric and has the most concise, standard mathematical representation. Therefore, it facilitates expansion into higher dimensional space. Compared with the traditional method which takes the most fine-grained points as input, the granular-ball computing takes the coarse-grained granular-balls as input, which is efficient, robust, and interpretable Xia et al. (2023). Granular-ball computing has expanded into various domains of artificial intelligence, as evidenced by works such as Xie et al. (2024); Quadir & Tanveer (2024); Zhang et al. (2023); Liu et al. (2024), etc. However, its application in optimization is relatively under-explored. Thus, this paper proposes a multi-granularity granular-ball optimization algorithm to explore this domain.

## 3 THE PROPOSED ALGORITHM

In this section, we present the multi-granularity optimization algorithm via granular-ball (GBO) for solving optimization problems (shown in Figure 2), which is composed of two modules: ((1) Multi-Granularity Solution Space Refinement: the solution space is refined from coarse-grained and fine-grained perspectives, respectively; (2) Granular-ball Exploration and Exploitation: the optimal solution is found through cooperative search among child granular-balls.

### 3.1 MULTI-GRANULARITY SOLUTION SPACE REFINEMENT

In this module, based on the "global precedence" cognitive law Chen (1982), a coarse-grained initial granular-ball is used to cover the solution space of the objective function. Then the sampling points operation is carried out inside the initial granular-ball to split many child granular-balls to refine the solution space.

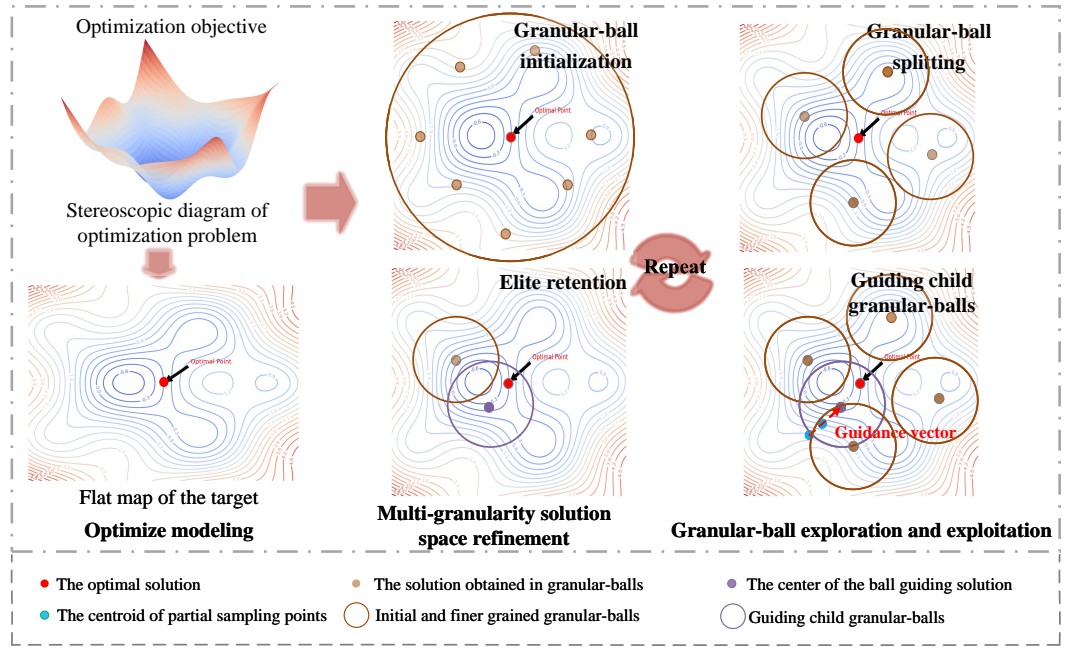

Figure 2: The framework of the proposed GBO. The figure illustrates the process of GBO using multi-granularity solution space refinement and granular-ball exploration and exploitation to solve optimization problems.

**Coarse-Grained Initialization.** In any dimension, a granular-ball needs only two data points to represent it: the center $c$ and the radius $r$. In a space of arbitrary dimensions, a granular-ball ($\mathcal{GB}$) is defined by its center vector $c$ and radius vector $r$. Given an initial granular-ball $\mathcal{GB}$, the center $c$ represents the position of the granular-ball in space and is a vector pointing to the center of the solution space. The radius $r$ is a vector where each component corresponds to half of the range in the respective dimension of the solution space. The initial granular-ball covers the entire solution space to ensure that no potential optimal solutions are overlooked.

The fitness value, as the only solution quality evaluation indicator in evolutionary computation, is indispensable for the algorithm. In this paper, due to the use of granular-balls instead of points to evaluate the search potential of a certain region in the solution space during the algorithm optimization process, the fitness of the search individuals in the algorithm, that is, the quality of the granular-balls, is redefined. The fitness value of the center of a granular-ball is taken as the quality of the granular-ball:

$$quality(\mathcal{GB}) = f(c). \tag{1}$$

**Fine-Grained Splitting.** When each granular-ball splits, the radius of the granular-ball is gradually reduced, and this process is also a transition from coarse-grained to fine-grained exploration. This strategy makes up for the shortcoming of the traditional evolutionary method that can only search on a single granularity and greatly improves the robustness of the algorithm to deal with problems of different complexity. In other words, the radii of the parent granular-ball and child granular-balls meet the following formula:

$$r_{t+1} = r_t \times \rho, t = 1, 2, .., t_{max} - 1 \tag{2}$$

where $r_{t+1}$ denotes the radii of child granular-balls in $t + 1$-th iteration, $r_t$ denotes the radius of parent granular-ball in $t$-th iteration, $\rho$ denotes the rate of radius reduction.

## 3.2 Granular-ball Exploration and Exploitation

Starting from an initial granular-ball that covers the solution space, each generation of granular-balls will undergo three processes: sampling points within the granular-ball, splitting, and selection. The radius of the granular-balls will gradually decrease, allowing for a more refined search of the solution space.

**Sampling points within the granular-ball.** The process for generating $n^*$ sampling points within a granular ball is as follows: First, $n^*$ uniform random numbers are generated in the interval $[0, 1]$, satisfying the condition $rand_k^j \sim U(0, 1)$, where $k = 1, 2, \ldots, n^*$ and $j = 1, 2, \ldots, D$. Here, $rand_k^j$ denotes the $k$-th random number in dimension $j$, and $D$ represents the dimension of the granular ball.

Then, the position of $k$-th sample point in dimension $j$ in the granular-ball is calculated based on random numbers:

$$x_k^j = lb_j + rand_k^j \times (ub_j - lb_j), k = 1, 2, \ldots n^*, j = 1, 2, \ldots, D, \tag{3}$$

where $lb_j$ is the lower bound of the $\mathcal{GB}$ in the $j$-th dimension, and $ub_j$ is the upper bound of the $\mathcal{GB}$ in the $j$-th dimension. After calculating the sampling points, randomly map $x_k^j$ that is out of range back into the defined domain.

Determining the number of sampling points for each particle sphere in each generation is a crucial process. The sampling points strategy can be mathematically expressed as $n = \frac{fes_{max}}{t_{max}}$, where $n$ represents the total number of sampling points in each iteration, $fes_{max}$ indicates the maximum number of fitness evaluations, and $t_{max}$ denotes the maximum number of iterations.

Thus, in each iteration, the number of sampling points for each granular ball, denoted as $\tilde{n}$, must satisfy the condition $\tilde{n} = \frac{n}{|G|}$, where $|G|$ represents the number of granular balls in that generation. This sampling strategy enhances the algorithm's adaptability across various problems.

Specifically, $\tilde{n}_1$ sampling points are first generated randomly for each parent granular-ball. Then, some child granular-balls are generated according to the no-overlapping generation strategy to maintain the diversity of granular-balls. Then, based on $\tilde{n}_1$ sampling points, $\tilde{n}_2$ guiding points are generated using the idea of gradient descent, and some guiding child granular-balls are generated at these points. $\tilde{n}_1$ and $\tilde{n}_2$ satisfy $\tilde{n} = \tilde{n}_1 + \tilde{n}_2$.

**No-overlaping child granular-balls generation strategy.** In this strategy, to maintain better exploration of a parent granular-ball, we aim for the child granular-balls formed by its splitting to be non-overlapping. Specifically, the centers of child granular-balls originating from the same parent granular-ball should not fall within the volume of another child granular-ball from that parent(Algorithm 1). For a parent granular-ball, initialize a set of child granular-balls $C_1$. Each time, randomly select a sampling point from the set of $\tilde{n}_1$ sampling points as the set $S$. If the sampling point is not inside any of the child granular-balls generated by the parent granular-ball, i.e., it satisfies the condition for all child granular-balls in the set $C_1$:

$$|x_k - c_i| \geq r_{t+1}, i = 1, 2, \ldots, |C_1| \tag{4}$$

where $x_k$ denotes the $k$-th sampling points, $c_i$ denotes the center of the $i$-th granular-ball, $|x_k - c_i|$ denotes the Euclidean distance between $x_k$ and $c_i$. This formula indicates that the center of a generated child granular-ball should not be inside the previously generated child granular-balls. If this condition is satisfied, a child granular-ball is generated with the sampling point as its center and a radius of $r_{t+1}$, and it is added to the set $C_1$.

**Guiding child granular-balls generation strategy.** The process of calculating the guiding vector to generate child granular-balls can be described as follows(Algorithm 2). Firstly, sort the fitness corresponding to the $\tilde{n}_1$ sampling points in ascending order. Secondly, select the top and bottom groups based on these sampling points. Calculate the centroids of two sets of sampling points as follows:

---

**Algorithm 1** No-overlaping child granular-balls generation strategy

---

**Input:** Sampling point set $S$ and granular-ball $\mathcal{GB}$.
**Output:** The non-overlapping child granular-balls $C_1$ in $\mathcal{GB}$.
1: $C_1 \leftarrow \{\}$;
2: Obtain the radius $r$ of granular-ball $\mathcal{GB}$;
3: $r \leftarrow r \times \rho$
4: **for** $bp$ **in** $S$ **do**
5:    **if** $bp$ is not within any granular-ball in $C_1$ **then**
6:       Generate a granular-ball centered at $bp$ with radius $r$ and add it to $C_1$;
7:    **end if**
8: **end for**
9: **return** $C_1$;

---

$$c_i^t = \frac{\sum_{j=1}^{\tilde{n}_1 \times \sigma} f(s_j)}{\tilde{n}_1 \times \sigma}, c_i^b = \frac{\sum_{j=\tilde{n}_1 - \tilde{n}_1 \times \sigma + 1}^{\tilde{n}_1} f(s_j)}{\tilde{n}_1 \times \sigma}, \tag{5}$$

where $s_j$ is the sampling point in $S$ with the $j$-th fitness value after sorting, $f(s_j)$ denotes the fitness of $s_j$, $\sigma$ is a hyper-parameter to control the number of sampling points in each group, $c_i^t$ and $c_i^b$ are the centroids of the two groups by the i-th granular-ball. Then, the guiding vector $\Delta_i$ is estimated by the difference between the two centroids in the i-th granular-ball:

$$\Delta_i = c_i^t - c_i^b. \tag{6}$$

Subsequently, the central position of $\tilde{n}_2$ guiding granular-balls are given:

$$\tilde{c} = c_i^t + \Delta_i \times w_i, \tag{7}$$

where $\tilde{c}$ denotes the center of $\tilde{n}_2$ guiding granular-balls, $w$ is the weight that controls the length of the guiding vector, and it satisfies a random uniform distribution in the interval [0.5, 1.5]. The guiding granular-ball strategy further improves the convergence speed of GBO.

---

**Algorithm 2** Guiding child granular-balls generation strategy

---

**Input:** Sampling point set $S$ and granular-ball $\mathcal{GB}$.
**Output:** The guiding child granular-balls in $\mathcal{GB}$.
1: Obtain the radius $r$ of granular-ball $\mathcal{GB}$;
2: Sort the fitness values of the sampling points in $S$ in ascending order;
3: $C_2 \leftarrow \{\}$;
4: $r \leftarrow r \times \rho$;
5: $c^t \leftarrow \frac{1}{\tilde{n}_1 \times \sigma} \sum_{j=1}^{\tilde{n}_1 \times \sigma} f(s_j)$;
6: $c^b \leftarrow \frac{1}{\tilde{n}_1 \times \sigma} \sum_{j=\tilde{n}_1 - \tilde{n}_1 \times \sigma + 1}^{\tilde{n}_1} f(s_j)$;
7: $\Delta \leftarrow c^t - c^b$;
8: **for** $i = 1$ **to** $\tilde{n}_2$ **do**
9:    Sample w from a specific distribution;
10:    $\tilde{c} \leftarrow c^t + \Delta \times w$;
11:    Generate a granular-ball centered at $\tilde{c}$ with radius $r$ and add it to $C_2$;
12: **end for**
13: **return** $C_2$

---

**Elite retention.** Typically, a generation of parent granular-balls produces many child granular-balls, which will waste lots of computational resources if they are all retained for the next iteration. Therefore, if the number of balls exceeds $N$, we sort the quality of all child granular-balls and select $N$ elite child granular-balls as the new generation granular-ball population for iterative search, otherwise all reserved.

**Iteration Loop.** The above multi-granularity design mechanisms for solution space and search methods work closely together to help GBO effectively find the optimal solution from coarse to fine granularity, making the algorithm capable of solving different optimization problems. Usually, after splitting to produce a new generation of granules, a new round of search will be conducted with them as the main body, and the search will be iteratively repeated until the consumption of computing resources is completed.

## 4 EXPERIMENTS

### 4.1 EXPERIMENT SETTINGS

**Benchmarks.** To verify the effectiveness of the GBO proposed in this paper, we conduct experiments on a commonly used CEC2013 benchmark Liang et al. (2013). There are 28 evaluation functions in CEC2013 benchmark, including 5 unimodal functions, 15 basic multimodal functions, and 8 composition functions. In addition, the CEC2011 real-world optimization problem set Das & Suganthan (2010) was used to verify the effectiveness of GBO in solving real-world optimization problems.

For a fair comparison, the number of given fitness evaluations for all algorithms is set to $10000 \times D$. This paper provides the mean errors (Mean) and standard deviations (Std.) obtained from 51 independent runs to assess the performance of all methods. Meanwhile, the specific experimental setup for GBO is: $\rho = 0.96$, $N = 30$, $t_{max} = 250$, $\sigma = 0.2$, $\tilde{n}_2 = 2$. We mainly presented the results of all algorithms in 30 dimensions for illustration purposes. In addition, for the sake of strict comparison, the Wilcoxon rank sum test was used at the significance level of $\alpha = 0.05$. Moreover, at a significance level of $\alpha = 0.05$, the Friedman test was used to comprehensively analyze the average rank (AR) obtained by each method on an overall problem set.

**Comparison Methods.** This paper first conducted a comprehensive comparison with classic evolutionary algorithms, including PSO Kennedy & Eberhart (1995), DE Qin et al. (2008), GA Holland (1992), ABC Karaboga et al. (2014), SHADE Tanabe & Fukunaga (2013) and LoTFWA Li & Tan (2017). Subsequently, GBO is compared with several popular variants of single objective global optimization algorithms including JADE Zhang & Sanderson (2009), MGFWA Meng & Tan (2024) (the SOTA variant of FWA), NSHADE Ghosh et al. (2022), LSHADE Tanabe & Fukunaga (2014) (CEC 2014's champion algorithm), PVADE dos Santos Coelho et al. (2013) and SPSO2011 Zambrano-Bigiarini et al. (2013) to further verify the performance of GBO.

### 4.2 EXPERIMENTAL RESULTS

**Overall Performance.** The experimental results are shown in Table 1 and Table 2. For each function, the optimal result is displayed in bold for emphasis. The mean errors followed by "+" indicate that GBO has good performance, the errors followed by "-" indicate that the comparison method has good performance, and the errors followed by "≈" indicate that the performance of GBO and comparison method is similar. It can be seen from the results in Table 1 that among the 28 evaluation functions, the performance of GBO exceeds that of comparison classic algorithms in 61%, 64%, 75%, 93%, 54%, and 71%, respectively. In addition, GBO has a mean rank of 2.52 across the 28 functions, which is far better than that of the comparison classic algorithms. As can be seen from Table 2, the performance of GBO is 57%, 71%, 54%, 57%, 75%, and 79% above the other six algorithms, respectively. In addition, the AR of GBO in 28 functions is 2.82, which is better than the comparison algorithms. The algorithm performs significantly better in testing complex functions compared to simpler ones, mainly due to the independent search between different granular-balls, resulting in good diversity.

**Ablation Studies.** We performed ablation experiments on the CEC2013 benchmark to examine the effects of the strategies described in the previous section on GBO. It mainly includes GBO-w/o guiding granular-balls. The results show that when GBO does not use a guiding strategy, the performance across 28 test functions is shown in Table 3.

This demonstrates that the guiding granular-balls strategy plays a crucial role in helping the model solve optimization problems and makes a significant contribution. This is because the centroids of mass guiding the granular-balls effectively dictate the subsequent search directions for the elite

Table 1: Comparison of GBO with several classic optimization algorithms in **30**-$D$.

| $f$ | GBO | | ABC | | DE | | GA | | PSO | | SHADE | | LoTFWA | |
|---|---|---|---|---|---|---|---|---|---|---|---|---|---|---|
| | Mean | Std. | Mean | Std. | Mean | Std. | Mean | Std. | Mean | Std. | Mean | Std. | Mean | Std. |
| $F_1$ | 3.25E-05 | 5.79E-06 | 4.55E-13- | 7.80E-14 | **0.00E+00-** | **0.00E+00** | 1.84E+00+ | 5.09E-01 | 2.77E+02+ | 5.05E+02 | **0.00E+00-** | **0.00E+00** | 1.27E-12- | 9.85E-13 |
| $F_2$ | 8.30E+05 | 4.46E+05 | 1.00E+07+ | 2.78E+06 | 3.88E+05- | 2.31E+05 | 2.29E+07+ | 1.08E+07 | 4.70E+06+ | 4.93E+06 | **1.26E+04-** | **1.05E+04** | 9.55E+05≈ | 4.25E+05 |
| $F_3$ | 3.37E+01 | 4.37E+00 | 7.04E+08+ | 4.86E+08 | **3.00E+01-** | **1.34E+02** | 5.62E+08+ | 5.26E+08 | 1.11E+10+ | 8.57E+09 | 2.53E+05+ | 1.26E+06 | 3.22E+07+ | 3.33E+07 |
| $F_4$ | 3.60E+04 | 9.53E+03 | 7.58E+04+ | 9.19E+03 | 1.54E+03- | 5.83E+02 | 1.40E+04- | 4.16E+03 | 3.36E+03- | 1.55E+03 | **1.25E-04-** | **3.21E-04** | 1.97E+03- | 7.33E+02 |
| $F_5$ | 3.53E-03 | 4.12E-04 | 7.80E-13- | 9.29E-14 | **9.14E-14-** | **4.51E-14** | 1.20E+00+ | 2.26E-01 | 4.19E+02+ | 6.24E+02 | 1.14E-13- | 1.26E-29 | 4.09E-03+ | 6.43E-04 |
| $F_6$ | 2.08E+01 | 1.89E+01 | 1.41E+01≈ | 4.59E+00 | 1.03E+01- | 4.79E+00 | 6.64E+01+ | 2.68E+01 | 7.24E+01+ | 4.00E+01 | **5.18E-01-** | **3.66E+00** | 1.56E+01≈ | 9.98E+00 |
| $F_7$ | **6.29E-02** | **8.95E-02** | 1.17E+02+ | 1.47E+01 | 3.73E+01+ | 7.13E-01 | 5.37E+01+ | 1.28E+01 | 1.83E+02+ | 1.08E+02 | 4.52E+00+ | 5.20E+00 | 5.36E+01+ | 1.37E+01 |
| $F_8$ | 2.09E+01 | 5.81E-02 | 2.09E+01≈ | 4.59E-02 | 2.09E+01≈ | 4.49E-02 | 2.10E+01+ | 4.80E-02 | 2.09E+01≈ | 6.19E-02 | **2.08E+01-** | **1.64E-01** | 2.09E+01≈ | 6.61E-02 |
| $F_9$ | **3.66E+00** | **1.93E+00** | 2.99E+01+ | 1.63E+00 | 3.76E+01+ | 4.48E+00 | 2.37E+01+ | 2.42E+00 | 3.48E+01+ | 3.03E+00 | 2.78E+01+ | 1.60E+00 | 1.71E+01+ | 2.07E+00 |
| $F_{10}$ | **1.66E-03** | **2.72E-03** | 1.84E+00+ | 4.33E-01 | 7.34E-03+ | 7.73E-03 | 3.08E+01+ | 1.37E+01 | 1.56E+02+ | 1.30E+02 | 6.81E-02+ | 3.18E-02 | 2.93E-02+ | 1.53E-02 |
| $F_{11}$ | 2.10E+01 | 4.96E+00 | 1.10E-13- | 2.08E-14 | 1.24E+02+ | 2.93E+01 | 1.78E+00- | 5.05E-01 | 2.67E+02+ | 5.85E+01 | **0.00E+00-** | **0.00E+00** | 8.78E+01+ | 1.46E+01 |
| $F_{12}$ | **2.13E+01** | **4.99E+00** | 2.73E+02+ | 3.97E+01 | 1.81E+02+ | 9.94E+00 | 7.90E+01+ | 1.79E+01 | 3.06E+02+ | 7.95E+01 | 2.26E+01≈ | 3.85E+00 | 8.68E+01+ | 1.67E+01 |
| $F_{13}$ | **3.85E+01** | **1.25E+01** | 3.10E+02+ | 3.02E+01 | 1.79E+02+ | 9.32E+00 | 1.57E+02+ | 3.11E+01 | 3.82E+02+ | 6.88E+01 | 4.99E+01+ | 1.27E+01 | 1.64E+02+ | 1.75E+01 |
| $F_{14}$ | 7.73E+02 | 2.74E+02 | 2.37E+00- | 1.46E+00 | 5.38E+03+ | 5.41E+02 | 1.12E+01- | 2.82E+00 | 3.98E+03+ | 8.49E+02 | **3.88E-02-** | **2.40E-02** | 2.78E+03+ | 2.80E+02 |
| $F_{15}$ | **7.88E+02** | **2.56E+02** | 3.85E+03+ | 2.98E+02 | 7.13E+03+ | 2.64E+02 | 4.25E+03+ | 6.34E+02 | 4.50E+03+ | 6.37E+02 | 3.36E+03+ | 3.12E+02 | 2.77E+03+ | 2.59E+02 |
| $F_{16}$ | **6.71E-03** | **2.57E-03** | 1.39E+00+ | 2.05E-01 | 2.48E+00+ | 2.79E-01 | 1.67E+00+ | 3.96E-01 | 1.48E+00+ | 3.76E-01 | 1.00E+00+ | 1.89E-01 | 1.59E-01+ | 5.19E-02 |
| $F_{17}$ | 4.84E+01 | 3.93E+00 | 3.05E+01- | 4.14E-02 | 1.85E+02+ | 1.56E+01 | 3.65E+01- | 1.02E+00 | 3.94E+02+ | 7.39E+01 | **3.04E+01-** | **1.38E-14** | 1.34E+02+ | 2.60E+01 |
| $F_{18}$ | **4.78E+01** | **3.65E+00** | 3.01E+02+ | 3.05E+01 | 2.11E+02+ | 9.98E+00 | 1.90E+02+ | 2.24E+01 | 4.10E+02+ | 7.85E+01 | 7.31E+01+ | 4.80E+00 | 1.44E+02+ | 2.21E+01 |
| $F_{19}$ | 3.00E+00 | 5.43E-01 | **4.50E-01-** | **1.18E-01** | 1.50E+01+ | 1.08E+00 | 2.00E+00- | 2.90E-01 | 6.33E+01+ | 1.63E+02 | 1.36E+00- | 1.11E-01 | 4.81E+00+ | 8.81E-01 |
| $F_{20}$ | **8.68E+00** | **6.81E-01** | 1.44E+01+ | 2.86E-01 | 1.23E+01+ | 2.69E-01 | 1.19E+01+ | 4.52E-01 | 1.41E+01+ | 5.72E-01 | 1.10E+01+ | 4.79E-01 | 1.30E+01+ | 1.14E+00 |
| $F_{21}$ | 2.48E+02 | 9.67E+01 | **1.78E+02-** | **3.16E+01** | 2.77E+02≈ | 6.18E+01 | 3.24E+02+ | 6.79E+01 | 3.50E+02+ | 1.10E+02 | 2.96E+02+ | 5.63E+01 | 2.02E+02- | 4.18E+01 |
| $F_{22}$ | 8.62E+02 | 2.21E+02 | **3.50E+01-** | **1.84E+01** | 5.24E+03+ | 8.11E+02 | 1.29E+02- | 4.09E+01 | 4.59E+03+ | 1.02E+03 | 8.50E+01- | 4.09E+01 | 3.31E+03+ | 4.09E+02 |
| $F_{23}$ | **8.57E+02** | **2.66E+02** | 4.80E+03+ | 4.81E+02 | 7.19E+03+ | 2.54E+02 | 4.44E+03+ | 6.21E+02 | 5.68E+03+ | 8.87E+02 | 3.61E+03+ | 4.39E+02 | 3.32E+03+ | 4.02E+02 |
| $F_{24}$ | **2.00E+02** | **1.45E-02** | 2.87E+02+ | 1.00E+01 | 2.25E+02+ | 1.26E+01 | 2.63E+02+ | 1.07E+01 | 3.11E+02+ | 1.07E+01 | 2.15E+02+ | 1.38E+01 | 2.42E+02+ | 7.46E+00 |
| $F_{25}$ | **2.12E+02** | **1.94E+01** | 3.06E+02+ | 4.65E+00 | 2.45E+02+ | 5.76E+00 | 2.80E+02+ | 9.25E+00 | 3.32E+02+ | 1.45E+01 | 2.79E+02+ | 9.04E+00 | 2.78E+02+ | 9.95E+00 |
| $F_{26}$ | 2.00E+02 | 2.56E-02 | 2.01E+02+ | 2.01E-01 | 2.03E+02+ | 1.80E+01 | 2.11E+02+ | 3.87E+01 | 3.17E+02+ | 8.98E+01 | 2.08E+02+ | 3.30E+01 | **2.00E+02≈** | **2.06E-02** |
| $F_{27}$ | **3.03E+02** | **1.80E-01** | 4.00E+02+ | 4.22E-01 | 5.87E+02+ | 1.16E+02 | 9.28E+02+ | 7.09E+01 | 1.25E+03+ | 9.24E+01 | 8.24E+02+ | 1.49E+02 | 7.80E+02+ | 5.91E+01 |
| $F_{28}$ | 3.00E+02 | 1.25E-02 | **2.11E+02-** | **7.73E+01** | 3.00E+02≈ | 5.68E-14 | 3.57E+02+ | 1.00E+01 | 1.85E+03+ | 1.14E+03 | 3.00E+02≈ | 1.19E-13 | 2.49E+02- | 8.71E+01 |
| $w/t/l$ | - | | 17/2/9 | | 19/3/6 | | 21/1/6 | | 26/1/1 | | 15/2/11 | | 20/4/4 | |
| $Rank$ | **2.52** | | 4.04 | | 4.16 | | 4.75 | | 6.43 | | 2.52 | | 3.59 | |

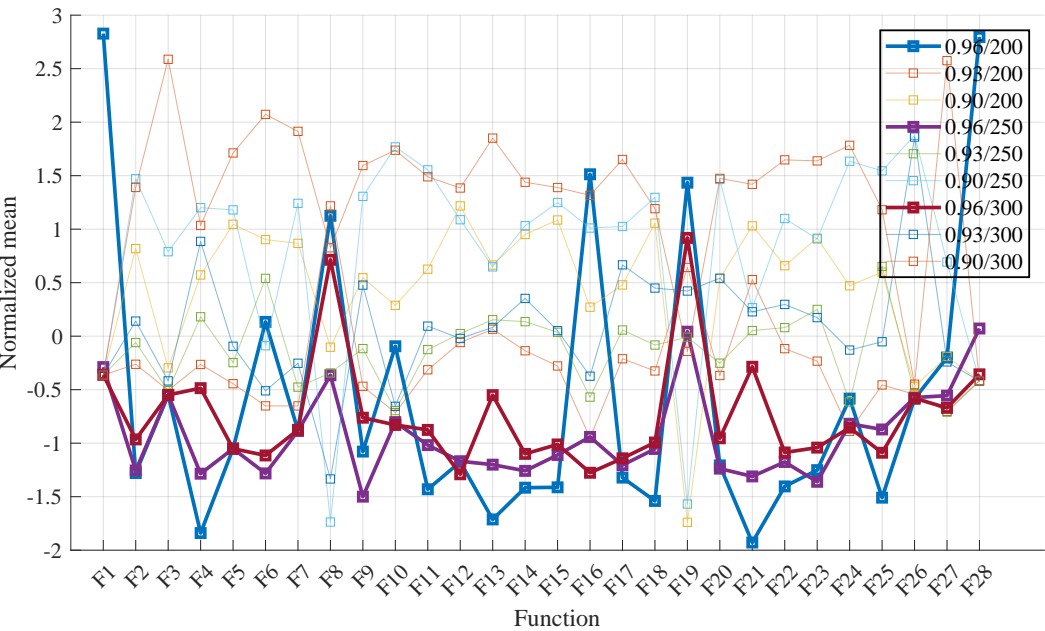

Figure 3: Comparison results of GBO based on 9 different parameter combinations on the 30-$D$ CEC2013 benchmark.

granular-balls within the solution space by aggregating information from high-quality sampling points. This mechanism not only enhances the efficiency of the search but also ensures a more precise approximation of the global optimal solution. Consequently, GBO is capable of rapidly identifying potentially favorable areas within a complex solution space and conducting thorough

Table 2: Comparison of GBO with several popular variants of single objective global optimization algorithms in **30**-$D$.

| $f$ | GBO Mean | GBO Std. | JADE Mean | JADE Std. | MGFWA Mean | MGFWA Std. | NSHADE Mean | NSHADE Std. | LSHADE Mean | LSHADE Std. | PVADE Mean | PVADE Std. | SPSO2011 Mean | SPSO2011 Std. |
|---|---|---|---|---|---|---|---|---|---|---|---|---|---|---|
| $F_1$ | 3.25E-05 | 5.79E-06 | **0.00E+00-** | **0.00E+00** | 3.57E-14- | 8.27E-14 | 2.23E-13- | 3.15E-14 | **0.00E+00-** | **0.00E+00** | **0.00E+00-** | **0.00E+00** | 8.92E-14- | 1.11E-13 |
| $F_2$ | 8.30E+05 | 4.46E+05 | **7.85E+03-** | **6.02E+03** | 1.41E+06+ | 4.95E+05 | 4.86E+04- | 2.97E+04 | 1.16E+04- | 8.62E+03 | 2.12E+06+ | 1.54E+06 | 2.31E+05 - | 8.80E+04 |
| $F_3$ | **3.37E+01 4.37E+00** | | 4.91E+05+ | 2.09E+06 | 6.42E+06+ | 9.46E+06 | 3.05E+06+ | 1.55E+07 | 7.62E+05+ | 2.14E+06 | 1.65E+03+ | 2.80E+03 | 1.89E+07+ | 1.97E+07 |
| $F_4$ | 3.60E+04 | 9.53E+03 | 3.44E+03- | 1.82E+03 | 1.22E+03- | 4.43E+02 | 2.18E+04- | 3.26E+04 | **2.03E-04-** | **4.49E-04** | 1.70E+04- | 2.82E+03 | 6.67E+03- | 1.67E+03 |
| $F_5$ | 3.53E-03 | 4.12E-04 | **1.09E-13-** | **2.21E-14** | 6.57E+03+ | 1.85E-03 | 2.76E-13- | 9.35E-14 | 1.14E-13- | 1.26E-29 | 1.40E-07- | 1.84E-07 | 9.34E-04- | 9.02E-05 |
| $F_6$ | 2.08E+01 | 1.89E+01 | **2.07E+00-** | **7.10E+00** | 1.49E+01- | 2.12E-01 | 6.05E+00- | 4.34E+00 | 2.77E+00- | 6.97E+00 | 8.29E+00- | 5.76E+00 | 2.13E+01+ | 2.18E+01 |
| $F_7$ | **6.29E-02 8.95E-02** | | 4.36E+00+ | 4.80E+00 | 2.56E+01+ | 8.42E+00 | 5.98E+01+ | 1.51E+01 | 4.84E+00+ | 4.55E+00 | 1.29E+00+ | 1.20E+00 | 1.82E+01+ | 9.34E+00 |
| $F_8$ | 2.09E+01 | 5.81E-02 | 2.09E+01≈ | 5.07E-02 | **2.08E+01-** | **5.94E-02** | 2.09E+01≈ | 5.26E-02 | 2.09E+01≈ | 5.51E-02 | 2.09E+01≈ | 4.77E-02 | 2.09E+01≈ | 7.00E-02 |
| $F_9$ | **3.66E+00 1.93E+00** | | 3.24E+01+ | 1.40E+00 | 9.98E+00+ | 1.82E+00 | 2.90E+01+ | 1.43E+00 | 2.77E+01+ | 1.84E+00 | 6.30E+00+ | 3.24E+00 | 2.60E+01+ | 5.07E+00 |
| $F_{10}$ | **1.66E-03 2.72E-03** | | 3.30E-02+ | 1.73E-02 | 2.53E-02+ | 2.00E-02 | 5.91E-02+ | 4.84E-02 | 7.60E-02+ | 5.36E-02 | 2.16E-02+ | 1.34E-02 | 1.96E-01+ | 8.93E-02 |
| $F_{11}$ | 2.10E+01 | 4.96E+00 | **0.00E+00-** | **0.00E+00** | 2.54E+01+ | 5.40E+00 | 5.80E-14- | 1.37E-14 | **0.00E+00-** | **0.00E+00** | 5.84E+01+ | 1.10E+01 | 5.43E+01+ | 2.73E+01 |
| $F_{12}$ | **2.13E+01 4.99E+00** | | 5.16E+01+ | 1.45E+01 | 2.65E+01+ | 5.70E+00 | 4.73E+01+ | 1.00E+01 | 2.42E+01+ | 3.26E+00 | 1.15E+02+ | 1.13E+01 | 4.11E+01+ | 1.21E+01 |
| $F_{13}$ | **3.85E+01 1.25E+01** | | 7.01E+01+ | 1.55E+01 | 5.01E+01+ | 1.31E+01 | 1.04E+02+ | 1.92E+01 | 4.79E+01+ | 9.99E+00 | 1.31E+02+ | 1.23E+01 | 8.91E+01+ | 1.92E+01 |
| $F_{14}$ | 7.73E+02 | 2.74E+02 | 5.10E-02- | 2.87E-02 | 2.39E+03+ | 3.58E+02 | 4.36E+00- | 1.41E+00 | **4.57E-02-** | **2.97E-02** | 3.20E+03+ | 4.34E+02 | 4.82E+03+ | 5.94E+02 |
| $F_{15}$ | **7.88E+02 2.56E+02** | | 6.54E+03+ | 3.86E+02 | 2.29E+03+ | 3.25E+02 | 3.17E+03+ | 3.46E+02 | 3.44E+03+ | 3.27E+02 | 5.61E+03+ | 3.15E+02 | 4.30E+03+ | 4.18E+02 |
| $F_{16}$ | **6.71E-03 2.57E-03** | | 2.37E+00+ | 2.82E-01 | 4.97E-02+ | 1.32E-02 | 8.00E-01+ | 1.48E-01 | 1.12E+00+ | 1.74E-01 | 2.39E+00+ | 2.63E-01 | 1.39E+00+ | 2.80E-01 |
| $F_{17}$ | 4.84E+01 | 3.93E+00 | **3.04E+01-** | **0.00E+00** | 5.60E+01+ | 4.88E+00 | 3.05E+01- | 3.01E-02 | **3.04E+01-** | **2.63E-14** | 1.02E+02+ | 1.16E+01 | 1.28E+02+ | 2.33E+01 |
| $F_{18}$ | **4.78E+01 3.65E+00** | | 1.70E+02+ | 9.47E+00 | 5.65E+01+ | 5.44E+00 | 8.75E+01+ | 8.04E+00 | 7.80E+01+ | 5.75E+00 | 1.82E+02+ | 1.19E+01 | 1.09E+02+ | 9.87E+00 |
| $F_{19}$ | 3.00E+00 | 5.43E-01 | 3.50E+00+ | 3.71E-01 | 2.39E+00+ | 4.13E-01 | 1.50E+01+ | 2.22E-01 | **1.46E+00-** | **1.21E-01** | 5.40E+00+ | 8.02E-01 | 5.66E+00+ | 2.93E+00 |
| $F_{20}$ | **8.68E+00 6.81E-01** | | 1.18E+01+ | 2.83E-01 | 1.27E+01+ | 1.29E+00 | 1.50E+01+ | 2.22E-01 | 1.11E+01+ | 3.84E-01 | 1.13E+01+ | 3.24E-01 | 1.07E+01+ | 5.75E-01 |
| $F_{21}$ | 2.48E+02 | 9.67E+01 | 2.83E+02≈ | 5.89E+01 | **2.11E+02-** | **3.00E+01** | 3.12E+02+ | 7.38E+01 | 2.98E+02+ | 5.99E+01 | 3.19E+02+ | 6.20E+01 | 3.19E+02+ | 5.73E+01 |
| $F_{22}$ | 8.62E+02 | 2.21E+02 | 2.01E+02- | 2.40E+02 | 2.78E+03+ | 4.07E+02 | **9.23E+01-** | **2.90E+01** | 1.06E+02- | 1.29E+01 | 2.50E+03+ | 3.82E+02 | 3.97E+03+ | 6.60E+02 |
| $F_{23}$ | **8.57E+02 2.66E+02** | | 6.51E+03+ | 3.93E+02 | 2.93E+03+ | 4.76E+02 | 3.98E+03+ | 3.74E+02 | 3.74E+03+ | 4.16E+02 | 5.81E+03+ | 4.99E+02 | 4.21E+03+ | 5.83E+02 |
| $F_{24}$ | **2.00E+02 1.45E-02** | | 2.42E+02+ | 2.40E+01 | 2.03E+02+ | 2.46E+00 | 2.29E+02+ | 5.82E+00 | 2.16E+02+ | 1.37E+01 | 2.02E+02+ | 1.38E+00 | 2.28E+02+ | 6.79E+00 |
| $F_{25}$ | **2.12E+02 1.94E+01** | | 2.85E+02+ | 7.81E+00 | 2.47E+02+ | 1.31E+01 | 2.91E+02+ | 1.87E+01 | 2.83E+02+ | 4.32E+00 | 2.30E+02+ | 2.06E+01 | 2.65E+02+ | 6.66E+00 |
| $F_{26}$ | **2.00E+02 2.56E-02** | | 2.35E+02+ | 6.37E+01 | **2.00E+02≈ 1.48E-02** | | **2.00E+02≈** | **3.52E-01** | 2.06E+02+ | 2.91E+01 | 2.18E+02≈ | 3.97E+01 | 2.17E+02+ | 4.38E+01 |
| $F_{27}$ | **3.03E+02 1.80E-01** | | 9.26E+02+ | 1.98E+02 | 3.44E+02+ | 2.96E+01 | 8.60E+02+ | 1.22E+02 | 8.70E+02+ | 1.17E+02 | 3.26E+02+ | 1.13E+01 | 5.80E+02+ | 5.55E+01 |
| $F_{28}$ | 3.00E+02 | 1.25E-02 | 3.00E+02≈ | 2.26E-13 | **2.96E+02- 2.77E+01** | | 2.96E+02- | 2.77E+01 | 3.00E+02≈ | 2.03E-13 | 3.00E+02 | 3.22E-05 | **2.96E+02-** | **2.77E+01** |
| $w/t/l$ | - | | 16/3/9 | | 20/1/7 | | 15/2/11 | | 16/2/10 | | 21/3/4 | | 22/1/5 | |
| $Rank$ | **2.82** | | 4.25 | | 3.61 | | 4.27 | | 3.18 | | 4.73 | | 5.14 | |

Table 3: Ablation Studies: GBO vs. GBO-w/o guiding granular-balls (w/t/l)

| Function Type | With Significance Level | Direct Value Comparison |
|---|---|---|
| Unimodal Functions | 3/2/0 | 5/0/0 |
| Basic Multimodal Functions | 6/8/1 | 14/0/1 |
| Composition Functions | 3/5/0 | 8/0/0 |
| All Functions | 12/15/1 | 27/0/1 |

explorations therein, thereby significantly improving both the quality of insights and the precision of the search.

**Hyper-Parameter Sensitivity Analysis.** We study the effect of different parameter combinations on the performance of GBO. Specifically, we use $\rho$ and $t_{max}$ for experiments in the ranges of 0.90, 0.93, 0.96, 200, 250, and 300, respectively, and the results are shown in Figure 3. The AR for the combination of these 9 parameters is depicted in Figure 4.

From our observation, GBO has the best performance when $\rho$ and $t_{max}$ are equal to 0.96 and 250, respectively. Under this combination of parameters, when GBO converges, the radius of the granular-ball becomes $10^{-5}$ of the initial radius. However, when $\rho$ and $t_{max}$ are equal to 0.90 and 300, respectively, the performance of GBO is the worst. Under this combination of

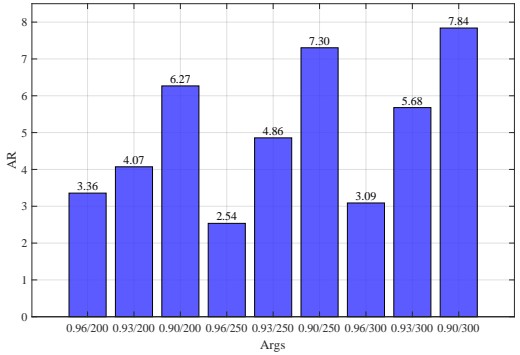

Figure 4: The AR results of GBO are based on 9 different parameter combinations.

parameters, when GBO converges, the radius of the granular-ball becomes $10^{-14}$ of the initial radius. When $\rho$ is smaller, the larger $t_{max}$ is, the less effective GBO is. However, performance does not always improve when $\rho$ is larger, and when $t_{max}$ is also increased. This phenomenon may be attributed to the fact that when the algorithm converges, the radius of the granular-ball should be within a suitable range, otherwise if the radius of the granular-ball is too small, then on the one hand, there is not much need to consume the number of fitness evaluations. On the other hand, approaching the local minimum too precisely may cause the algorithm to fall into the local minimum, which will negatively affect the optimization performance of the algorithm.

**Case Study.** We apply GBO to solve the transmission network expansion planning (TNEP) problems. The simple TNEP Abraham & Das (2010); Silva et al. (2005) without safety constraints determines the new line set to be built, minimizes the cost of the expansion plan, and does not generate overloads within the planned range. The detailed modeling is taken from the CEC2011 test set Das & Suganthan (2010). We compare the results with the champion algorithm GA-CMP Elsayed et al. (2011b) and the third-place algorithm SAMODE El-sayed et al. (2011a) of the CEC2011 competition. The parameters of GBO are $N = 10$, $\rho = 0.94$, the solution is performed with $t_{max}$ of 150, and $fes_{max}$ of each algorithm is 150,000. The results are shown in Table 3, from which it can be seen that GBO has shown great advantages in this practical problem.

Table 4: Detailed comparison between SAMODE, GA-MPC, and GBO on CEC2011 real-world problem.

| $fes$ | Metric | SAMODE | GA-MPC | GBO |
|---|---|---|---|---|
| | Best | 8.21E-01 | 7.75E-01 | **6.39E-01** |
| | Median | 1.27E+00 | 1.74E+00 | **7.92E-01** |
| 50000 | Worst | 1.70E+00 | 1.92E+00 | **1.06E+00** |
| | Mean | 1.29E+00 | 1.62E+00 | **8.07E-01** |
| | Std. | 1.93E-01 | 3.24E-01 | 8.81E-03 |
| | Best | 5.08E-01 | 5.08E-01 | **5.00E-01** |
| | Median | 9.99E-01 | 7.95E-01 | **5.58E-01** |
| 100000 | Worst | 1.33E+00 | 1.68E+00 | **7.69E-01** |
| | Mean | 9.73E-01 | 8.58E-01 | **5.80E-01** |
| | Std. | 1.79E-01 | 2.73E-01 | 6.54E-03 |
| | Best | **5.00E-01** | **5.00E-01** | **5.00E-01** |
| | Median | 8.40E-01 | 7.58E-01 | **5.47E-01** |
| 150000 | Worst | 9.94E-01 | 9.33E-01 | **7.57E-01** |
| | Mean | 8.17E-01 | 7.48E-01 | 5.73E-01 |
| | Std. | 1.19E-01 | 1.25E-01 | **6.10E-03** |

## 5 CONCLUSION

In this paper, we propose a multi-granularity optimization algorithm (GBO) via granular-ball. Aiming at the multi-granularity of the solution space, GBO uses a splitting mechanism to cover the solution space, carries out a global search from coarse-grained to fine-grained, and finds the optimal solution through synergistic search between granular-balls. It replaces the traditional point-based iterative and regional search methods, allowing for a more comprehensive consideration of the complexity and uniqueness of the solution space. Experiments on the CEC2013 benchmark and CEC2011 real-world problems confirm the superiority of GBO. However, we have yet to design a more adaptive method for the radius of each generation of granular-balls, allowing their offspring to exhibit varying granularities to enhance the efficiency of the GBO method. To address this issue, we plan to adopt more effective strategies in our future work.

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

## A  BENCHMARK FUNCTIONS

The twenty-eight benchmark functions from CEC2013 are used to evaluate the performance of the GBO algorithm Liang et al. (2013). These functions include five unimodal functions, fifteen multi-modal functions, and eight composite functions, as shown in Table 3. These functions possess numerous local minima and maxima, with the global optimum randomly distributed within the range of [-100, 100]. Additionally, the integration of orthogonal (rotated) matrices into these functions further enhances their complexity. In summary, these test functions are highly complex, presenting significant challenges for the performance assessment of the algorithm.

Table 5: Benchmark functions used in CEC2013

|  | No. | Functions | $f_i^* = f_i(x^*)$ |
|---|---|---|---|
| Unimodal Functions | 1 | Sphere Function | -1400 |
| | 2 | Rotated High Conditioned Elliptic Function | -1300 |
| | 3 | Rotated Bent Cigar Function | -1200 |
| | 4 | Rotated Discus Function | -1100 |
| | 5 | Different Powers Function | -1000 |
| Basic Multimodal Functions | 6 | Rotated Rosenbrock's Function | -900 |
| | 7 | Rotated Schaffers F7 Function | -800 |
| | 8 | Rotated Ackley's Function | -700 |
| | 9 | Rotated Weierstrass Function | -600 |
| | 10 | Rotated Griewank's Function | -500 |
| | 11 | Rastrigin's Function | -400 |
| | 12 | Rotated Rastrigin's Function | -300 |
| | 13 | Non-Continuous Rotated Rastrigin's Function | -200 |
| | 14 | Schwefel's Function | -100 |
| | 15 | Rotated Schwefel's Function | 100 |
| | 16 | Rotated Katsuura Function | 200 |
| | 17 | Lunacek Bi_Rastrigin Function | 300 |
| | 18 | Rotated Lunacek Bi_Rastrigin Function | 400 |
| | 19 | Expanded Griewank's plus Rosenbrock's Function | 500 |
| | 20 | Expanded Scaffer's F6 Function | 600 |
| Composition Functions | 21 | Composition Function 1 (n=5, Rotated) | 700 |
| | 22 | Composition Function 2 (n=3, Unrotated) | 800 |
| | 23 | Composition Function 3 (n=3, Rotated) | 900 |
| | 24 | Composition Function 4 (n=3, Rotated) | 1000 |
| | 25 | Composition Function 5 (n=3, Rotated) | 1100 |
| | 26 | Composition Function 6 (n=5, Rotated) | 1200 |
| | 27 | Composition Function 7 (n=5, Rotated) | 1300 |
| | 28 | Composition Function 8 (n=5, Rotated) | 1400 |
| Search Range: $[-100, 100]^D$ | | | |

# B   THE PSEUDO-CODE OF GBO.

The complete pseudocode of the GBO algorithm is as follows. The algorithm framework is simple and easy to implement.

---

**Algorithm 3** The multi-granularity optimization algorithm via granular-ball (GBO)

---

**Input:** The optimization objective $f$ and maximum number of iterations $fes_{max}$.
**Output:** The best fitness of $f^*$ and its corresponding solution position $bp^*$.

1: $G \leftarrow \{\}$;
2: $n \leftarrow \frac{fes_{max}}{t_{max}}$;
3: Initialize a granular-ball that covers the solution space and add it to $G$;
4: **for** $j = 1$ **to** $t_{\max}$ **do**
5:   $\tilde{n} \leftarrow \frac{n}{|G|}$;
6:   $\tilde{n}_1 \leftarrow \tilde{n}$ - $\tilde{n}_2$;
7:   $G_{child} \leftarrow \{\}$ ;
8:   Calculate the fitness values of the sampling point set $S$;
9:   **for** $i = 1$ **to** $|G|$ **do**
10:     Generate $\tilde{n}_1$ sampling points within $\mathcal{GB}_i$ as $S_i$;
11:     Perform random mapping on $S_i$;
12:     $C_1 \leftarrow$ Alg.1$(S_i, \mathcal{GB}_i)$;
13:     $C_2 \leftarrow$ Alg.2$(S_i, \mathcal{GB}_i)$;
14:     $G_{child} \leftarrow G_{child} \cup C_1 \cup C_2$;
15:   **end for**
16:   Sort the child granular-balls in $G_{child}$ in ascending order of mass;
17:   Select min $\{|G|, N\}$ elite granular-balls as $G$ in $G_{child}$;
18:   Update $f^*$ and $bp^*$;
19: **end for**
20: **return** $f^*$ and $bp^*$;

---

# C   COMPARE THE PARAMETER SETTINGS OF THE ALGORITHMS.

In the experimental section, the parameters used for the comparison algorithms are as follows: the population size is 100 for all algorithms except for the variants of the FWA algorithm, and all other parameters are set to their optimal values.

Table 6: The parameter setting of comparison algorithms.

| Algorithms | Parameters | Values |
|---|---|---|
| PSO | $N, c_1, c_2, w$ | 100,2, 2, 0.9-0.4 |
| DE | $N, F, CR$ | 100, 0.5, 0.9 |
| GA | $N, MR, CR$ | 100, 0.1, 0.8 |
| ABC | $N, Limit, sn$ | 100, 200, 1 |
| SHADE | $N, H, F, CR$ | 100, 100, 0.5, 0.5 |
| LoTFWA | $fw_{size}, sp_{size}, init_{amp}, gm_{ratio}$ | 5, 300, 200, 0.2 |
| JADE | $N, F, CR, pt, ap$ | 100,0.5, 0.5, 0.1, 0.1 |
| MGFWA | $fw_{size}, sp_{size}, init_{amp}, gm_{ratio}, parameter_N, parameter_b$ | 5, 300, 200, 0.2, 10, 1.5 |
| NSHADE | $N, F, CR$ | 100, 0.5, 0.5 |
| LSHADE | $N, F, CR$ | 100, 0.5, 0.5 |
| PVADE | $N$ | 100 |
| SPSO2011 | $N, w, c_1, c_2$ | 100, $\frac{1}{2 \times \ln 2}$, 0.5+$\ln 2$, $0.5 + \ln 2$ |

