# OpenReview forum: "GBO: A Multi-Granularity Optimization Algorithm via Granular-ball for Continuous Problems"
_ICLR.cc/2025/Conference — ICLR 2025 Conference Withdrawn Submission_

### Official Review · Reviewer_f6wW · 2024-10-28

**Soundness:** 2
**Presentation:** 2
**Contribution:** 1
**Rating:** 3
**Confidence:** 5

**Summary:**

This paper introduces a multi-granularity version of the GBO algorithm, which falls under the category of evolutionary algorithms/swarm intelligence optimization algorithms. Simply put, it focuses on multi-granularity during sampling (generating sub-granular ball spaces and ensuring non-intersection), rather than sampling all the samples at once, which improves efficiency. It also introduces a guiding vector (a method similar to gradients for search). The paper also mentions that Table 3 describes a very important operation, and the algorithm has achieved certain effects on some basic test sets, but we can see that the performance is not particularly excellent on CEC2013 benchmark or a single task on CEC2011 project problem.

**Strengths:**

This paper's writing is clear and easy to understand. Many mechanisms from various optimization algorithms have been integrated together.

**Weaknesses:**

There are several major issues as follows:
1、The level of innovation in this paper is relatively low. The idea of multi-granularity sampling or multi-level sampling is a very conventional concept that has been mentioned in many works, lacking in novelty. (you can  refer to EDEFWA, almost the same)
2、The guiding vector is even more traditional, beginning in GFWA almost a decade ago. This cannot be considered the authors' own contribution. Including the LoTFWA and MGFWA cited in the paper, it can be said that the guiding vector is a very important mechanism in the fireworks algorithm.
3、The experimental results are not particularly excellent, and there is a lack of one-on-one statistical testing. Relying solely on the average rank is quite weak.
4、There is not much theoretical support. For example, the paper does not explain why this algorithm performs better on certain problems. According to the No Free Lunch (NFL) theorem, there may be other performances that are compromised, and this content is not mentioned in the paper.
MGFWA:https://linkinghub.elsevier.com/retrieve/pii/S2210650223002304
LoTFWA:https://ieeexplore.ieee.org/document/8239679/
GFWA:https://ieeexplore.ieee.org/document/7508443
EDEFWA：https://ieeexplore.ieee.org/document/9504974

**Questions:**

(1)Why did the experiment only test one task on cec2011 project problem.  I think it may not be enough.
(2)You didn't mention the mechanism is from FWA variant and made it as your contribution.

---

### Official Review · Reviewer_ugx8 · 2024-11-02

**Soundness:** 2
**Presentation:** 1
**Contribution:** 1
**Rating:** 3
**Confidence:** 5

**Summary:**

This paper proposes a heuristic black-box optimization algorithm based on Granular-balls, which iteratively sample solutions from a uniform distribution inside the balls. An empirical study verifies the effectiveness of the proposed method.

**Strengths:**

* An executable code is provided.
* Generally, this paper is easy to follow.
* An empirical study is conducted, which shows that the proposed method outperforms the baselines.
* Non-parametric tests are included in the empirical study.

**Weaknesses:**

1. As proposed in this paper, the concept of "granular characteristic" is one of the key highlights. However, this concept is still a little bit confusing to me. In L65, it says "regions proximate to the global optimum are characterized by finer granularity". So it seems that the so-called "granular characteristic" is simply the distance to the global optimum. During the optimization process, the position of the global optimum is unknown; therefore, it remains unclear how the proposed method has successfully modeled the "granular characteristic".
2. I think the proposed method can be categorized as an Estimation-of-Distribution algorithm (EDA) [1], which has been widely studied for decades. However, this paper does not include EDAs in the literature review or compare them in the experiments. Similar to GBO proposed in this paper, EDAs also model solutions from a distribution (or several independent distributions), and the distribution evolves from coarse to fine. One of the common practices is to sample solutions from a Gaussian distribution [2], which seems more reasonable than sampling from a uniform distribution in a ball.
3. The empirical study can be further improved. I suggest reporting the convergence curves of the baselines in all the test instances. Furthermore, many components in GBO are not included in the ablation study.
4. According to Fig. 4, the performance of GBO relies heavily on parameter tuning, which might be a significant limitation.

References

[1] Hauschild, Mark, and Martin Pelikan. "An introduction and survey of estimation of distribution algorithms." Swarm and evolutionary computation 1.3 (2011): 111-128.

[2] Hansen, Nikolaus. "The CMA evolution strategy: A tutorial." arXiv preprint arXiv:1604.00772 (2016).

**Questions:**

Please refer to "Weaknesses".

---

### Official Review · Reviewer_jYSg · 2024-11-02

**Soundness:** 2
**Presentation:** 2
**Contribution:** 2
**Rating:** 5
**Confidence:** 3

**Summary:**

The paper proposes an heuristic optimization method for continuous problems based on granular-balls. The algorithm through the balls partitions the search space, and the granularity of the balls changes through the optimization going from coarse to a fine.

It can be seen as an evolutionary algorithm that instead of sampling and exploring the search space through points, these are replaced by d-dimensional balls of radius r that cover sub-regions of it, and using a temperature like parameter, the radius r is slowly decreased to go from a coarse search to a fine one.

**Strengths:**

Methods to split the search space are not uncommon, for example niching in genetic algorithms, the proposed scheme via granular balls seems to exploit well the fact that solutions close in parameter space can be encoded as being generated inside a d-dimensional ball representation. This also acts as a divide and conquer strategy that splits the search space into tractable regions that now are represented by a single fitness value.

The figures help visualizing the partitioning of the search space, and how the method determines where it should generate the new balls for the following iteration. The introduction gives an overview of evolutionary methods and the relevance of heuristic search when problems are not suited for gradient based methods.
Experiments seem comprehensive covering the baseline methods like an standard Genetic Algorithm, Particle Swarm Optimization or Differential Evolution, Artificial Bee Colony, to more recent ones based on the Differential Evolution scheme.

**Weaknesses:**

The method determines the fitness of a ball by the solution at the center, which does not seem to convey properly the value of a particular region. Is unclear what happens when the center is not the best possible choice.

New child balls are generated using the points sampled inside their parent ball at random and using the center to generate them inside the volume of the parent but with no overlaps. Which should help exploration and hopefully hit a better solution inside the region, but it seems instead of this random approach a smarter scheme could be made if each ball used the information of the points inside it.

The method could hit the optimum faster if the fitness of all solutions inside a region informed the algorithm how the region could be split or which are should be focused next.

Figure 2 needs a way to convey the order we should follow to understand the method. It seems the right order is top left, top right, bottom right and finally bottom left.

**Questions:**

1- The fitness of the ball is the fitness of the solution at that point. Is this the best choice? wouldn't a summary statistic like the mean be more accurate to represent the value of sampling that ball? For an example on how this could be achieved, CMA-ES or any ES method that does the search using a distribution to sample the space could be modified with the granular ball technique.

2- In the paragraph that starts at line 245, there is a mention that the guiding points are generated using the idea of gradient descent, but this is not clearly explained.

3- One possible suggestion to the radius adjustment problem could be to implement some annealing similarly to the one done to the learning rate.

---

### Official Review · Reviewer_6zGZ · 2024-11-04

**Soundness:** 3
**Presentation:** 3
**Contribution:** 2
**Rating:** 5
**Confidence:** 3

**Summary:**

This paper proposes a new multi-granularity optimization algorithm (GBO) that characterizes and searches the solution space from coarse to fine using granular-balls,which effectively solved the continuous optimization problem. The authors have validated the effectiveness of the GBO algorithm on the CEC2013 benchmark and CEC2011 real-world problems, demonstrating its superiority over various classic evolutionary algorithms.

**Strengths:**

1. The paper introduces an innovative approach to multi-granularity optimization problems through the introduction of granular-balls, providing a new perspective on solving continuous optimization problems.
2. The experimental part is well-designed, using the CEC2013 and CEC2011 datasets, and the results show the effectiveness of the GBO algorithm.
3. The results are analyzed in detail, and the comparative experiments highlight the advantages of the GBO algorithm.

**Weaknesses:**

1. Although the GBO algorithm performs well on low-dimensional problems, its performance and efficiency on high-dimensional such as BBOB-Largescale [1] still need further verification.
2. The paper would benefit from a formal analysis of convergence properties and robustness against local optima, which are critical in validating the method’s scalability.


[1] Elhara O, Varelas K, Nguyen D, et al. COCO: the large scale black-box optimization benchmarking (BBOB-largescale) test suite[J]. arXiv preprint arXiv:1903.06396, 2019.

**Questions:**

1. How does GBO perform in very high-dimensional spaces or when the solution space complexity increases significantly? For example, are there experimental results on high-dimensional benchmarks like BBOB-LargeScale [1].
2. How does GBO handle noise in the fitness function, and how robust is it to variations in the initial conditions?
3. The paper would benefit from a formal analysis of convergence properties and robustness against local optima, such as convergence properties and robustness against local optima mentioned in weakness 2.

---

### Note · Authors · 2024-11-27

I have read and agree with the venue's withdrawal policy on behalf of myself and my co-authors.